# Synergistic Effect of a Pleuromutilin Derivative with Tetracycline against *Streptococcus suis* In Vitro and in the Neutropenic Thigh Infection Model

**DOI:** 10.3390/molecules25153522

**Published:** 2020-08-01

**Authors:** Fang Chen, Meng-Chao Wei, Yi-Dan Luo, Zhen Jin, You-Zhi Tang

**Affiliations:** 1Guangdong Provincial Key Laboratory of Veterinary Pharmaceutics Development and Safety Evaluation, College of Veterinary Medicine, South China Agricultural University, No. 483 Wushan Road, Tianhe District, Guangzhou 510642, China; fox280259306@163.com (F.C.); 18826487461@163.com (M.-C.W.); shdaila521@163.com (Y.-D.L.); 2Guangdong Laboratory for Lingnan Modern Agriculture, Guangzhou 510642, China

**Keywords:** *Streptococcus suis*, tetracycline, combination therapy, checkerboard assays, time-killing assays, thigh infection model

## Abstract

Tetracycline (TET) has been widely used in the treatment of *Streptococcus suis* (*S. suis*) infection. However, it was found that the efficacy of many antibiotics in *S. suis* decreased significantly, especially tetracycline. In this study, GML-12 (a novel pleuromutilin derivative) was used in combination with TET against 12 *S. suis* isolates. In the checkerboard assay, the TET/GML-12 combination exhibited synergistic and additive effects against *S. suis* isolates (*n* = 12). In vitro time-killing assays and in vivo therapeutic experiments were used to confirm the synergistic effect of the TET/GML-12 combination against *S. suis* strains screened based on an FICI ≤ 0.5. In time-killing assays, the TET/GML-12 combination showed a synergistic effect or an additive effect against three isolates with a bacterial reduction of over 2.4-log_10_ CFU/mL compared with the most active monotherapy. Additionally, the TET/GML-12 combination displayed potent antimicrobial activity against four isolates in a mouse thigh infection model. These results suggest that the TET/GML-12 combination may be a potential therapeutic strategy for *S. suis* infection.

## 1. Introduction

*Streptococcus suis* (*S. suis*) is a significant zoonotic pathogen that causes a variety of diseases, ranging from meningitis to serious blood infections, and an impact on human health with deafness [1,2]. People who usually come in contact with infected pigs or pork-derived products are at risk, such as pig farmers, slaughterhouse workers and veterinarians [3]. In China, two large outbreaks of human *S. suis* infection occurred in 1998 and 2005 [4,5]. *S. suis* infections in human have been reported from over 30 countries, and the number of cases has continued to rise in recent decades [6]. It poses a significant threat to public health and causes considerable economic losses in the pig industry [7]. Various types of vaccines have been developed to prevent *S. suis* strains infection, but their preventive efficacy was not stable [8,9]. Considering the economic benefits and therapeutic effects, the use of antibiotics was the first choice against *S. suis* [10].

Antibiotics including aminoglycoside, β-lactam, macrolides, lincomycin and tetracycline are usually used for the treatment of *S. suis* infection [11]. Tetracycline (TET), as a broad-spectrum antibiotic, is widely used all over the world due to its easy absorption, its low price and its minimum side effects [12]. However, with the increase in ineffective uses of TET, the resistance rate of *S. suis* to tetracycline increased rapidly and a high occurrence of TET resistance was found in China (99.1%) [13,14]. Although the development of antibiotics can meet the challenge of antibiotic resistance, there are some problems still existing such as it takes a long time and a high cost to bring new drugs to clinical use [15,16]. In such a scenario, the diversified use of antibiotics is more practical than the development of new drugs. At present, a series of reports have shown that successful combination therapy can effectively reduce the dosage of traditional antibiotics [17], with a greater curative effect and fewer side effects [18]. Restoring the sensitivity of previously resistant bacteria to old antibiotics may be the most powerful effect of a synergistic combination [17,19].

It has been reported that the combination of a pleuromutilin derivative and doxycycline displayed a strong synergistic effect against multi-drug-resistant *Acinetobacter baumannii* [20]. It is well established that the antibacterial mechanisms of pleuromutilin derivatives are to inhibit protein biosynthesis by blocking the peptidyl transferase center (PTC) of the 50S ribosomal protein L3 [21]. GML-12 was also found to form a hydrogen bond with the 50S ribosome in a previous docking study [22]. TET inhibits protein synthesis by binding to the A position of the bacterial ribosomal 30S subunit, which is different from the site where GML-12 acts [23]. Theoretically, compared with monotherapy, the combination of TET/GML-12 can improve clinical efficacy and broaden the spectrum of antibacterial activity. Thus, we speculate that tetracyclines combined with pleuromutilin derivatives may be an effective strategy for the treatment of *S. suis* infection. A series of novel pleuromutilin derivatives have been reported in our previous work [22], in which one derivative, 22-(4-(2-(4-Nitrophenyl-piperazin-1-yl)-acetyl)-piperazin-1-yl)-22-deoxypleuromutilin (GML-12), displayed potent antibacterial activity against MRSA both in vitro and in vivo. In this study, we have investigated the in vitro and in vivo activity of GML-12, TET and their combination against *S. suis* strains, and evaluated whether there was a synergistic effect between the two antibiotics.

## 2. Results

### 2.1. In Vitro Antibacterial Activity

#### 2.1.1. MIC and Checkerboard Assay

The results from MIC and checkerboard assays for all *S. suis* strains are shown in Table 1. Except for ATCC43765 and S1-2, which were susceptible and intermediate to TET, respectively, the other *S. suis* strains were resistant to TET (MIC, 32–128 μg/mL). The MIC values of GML-12 against eight *S. suis* strains were less than 1 µg/mL. GML-12 exhibited more significant antibacterial activity than TET, with an MIC range of 0.0625–8 µg/mL. The MIC_50_ value of TET and GML-12 was 64 and 0.125 μg/mL, respectively. The results from checkerboard assays suggested synergistic or additive for all *S. suis* strains, with the lower FICI values for ATCC 43765, S40, S11 and SNJ-5 ranging from 0.375 to 0.5. There were no antagonistic interactions or instances of indifference between TET and GML-12. The synergistic effect of antibiotics caused a reduction ranging between 4- to 8-fold for the MIC of TET and 4-fold for the effective concentration of GML-12. When an additive effect was exhibited, the MIC values of TET and GML-12 were also reduced from 2- to 8-fold and 4-fold, respectively.

#### 2.1.2. Time-Killing Assays of Synergistic Combinations

The time-killing assays of GML-12 and TET, alone and in combination, against four *S. suis* strains (ATCC 43765, S40, SNJ-5, S11) are presented in Figure 1. The combination of TET (1 × MIC) and GML-12 (1 × MIC) resulted in an obvious drop of 3.84-log_10_ CFU/mL and 3.55-log_10_ CFU/mL compared with the most active single drug in the bacterial counts over 24 h of exposure in the ATCC43765 (Figure 1A) inoculum and S11 (Figure 1D) inoculum, respectively. As described above, a bactericidal effect was defined as a ≥ 3-log10 reduction in CFU/mL compared with the initial inoculum. For *S. suis* SNJ-5 (Figure 1C), the combination of TET at 1 × MIC plus GML-12 at 2 × MIC showed a synergistic effect, with a drop of 2.48-log_10_ CFU/mL. TET (2 × MIC) alone or combined with GML-12 (1 × MIC, 2 × MIC) killed all *S. suis* SNJ-5 after 24 h. It seems that the interactions between them were indifferent according to the reduction of lower than 1-log_10_ CFU/mL. Since the bacterial density of the combination therapy and single therapy was 0-log_10_ CFU/mL, which was the minimum value that could be measured by the time-killing assay, the result only showed that the interaction between the TET and GML-12 was not an antagonistic effect. An additive effect was observed when the combination of TET (2 × MIC) and GML-12 (2 × MIC) resulted in a reduction of 1.31-log_10_ CFU/mL compared with TET (2 × MIC) against *S. suis* S40 (Figure 1B). There were no antagonistic effects between TET and GML-12 in all time-killing assays.

### 2.2. In Vivo Assessment

#### 2.2.1. Acute Toxicity of GML-12 in Mice

In terms of the intravenous (IV) and oral (PO) administration routes, GML-12 did not show any sign of stress or abnormal behavior when injected at 20 mg/kg. In addition, no mice died up to two weeks post-treatment (Figure 2).

#### 2.2.2. In Vivo Synergistic Effects of GML-12/TET Combination

The mice thigh infection model was established to evaluate the synergetic effect of TET and GML-12 against four *S. suis* strains (ATCC 43765, S40, SNJ-5, S11) in vivo. The change in viable bacterial counts in mouse thighs were determined by colony counting. By comparing the bacterial density of the combined treatment with that of the control group, it was observed that the TET/GML-12 combination displayed potent antimicrobial activity against ATCC 43,765 (*p* ≤ 0.0001), S40 (*p* ≤ 0.0001), SNJ-5 (*p* ≤ 0.0001) and S11 (*p* = 0.0016). When comparing the growth of ATCC 43,765 (Figure 3A) in a thigh infection model between monotherapy (TET or GML-12) and the TET/GML-12 combination, there was a statistically significant difference (TET: 8.303 ± 0.1928 log10 CFU/g, GML-12: 7.651 ± 0.1642 log10 CFU/g, TET/GML-12 combination: 6.087 ± 0.4526 log10 CFU/g, *n* = 6, *p* ≤ 0.0087), showing a synergistic effect. The minimal reduction of *S. suis* S40 (Figure 3B) and *S. suis* SNJ-5 (Figure 3C) was by 0.7638 ± 0.1784 log10 CFU/g (*p* = 0.016) and 2.234 ± 0.5238 log10 CFU/g (*p* = 0.0016), respectively, between a combination of TET and GML-12 with monotherapy (TET or GML-12), which also indicated a synergistic effect. In the thigh infection model with *S. suis* S11 (Figure 3D), there was no statistically significant difference in bacterial density between monotherapy (TET or GML-12) and the TET/GML-12 combination (TET/GML-12 combination: 7.665 ±0.1505 vs. TET: 7.765 ± 0.1701 log10 CFU/g, *p* = 0.6692; TET/GML-12 combination vs. GML-12: 8.005 ± 0.1953 log10 CFU/g, *p* = 0.198). GML-12 at a dose of 16 mg/kg was significantly more effective (*p* < 0.05) at 24h than was TET at a dose of 256 mg/kg (*p* = 0.0016) in the infection model with *S. suis* SNJ-5. For other strains (ATCC43765, S11 and S40), there were no statistically significant differences in antibacterial activity between TET treatment and GML-12 treatment. In all mice infection models, the therapeutic dose of GML-12 was 4 to 512 times lower than the dose of tetracycline.

## 3. Discussion

*S. suis* was a typical bacterial pathogen in modern pig production [26], which is presently considered an emerging multidrug-resistant (MDR) bacteria [27]. High resistance levels were found for TET and tiamulin (TIA) during the period 2004–2015 [28]. In this study, most *S. suis* strains were resistant to TET (MIC_50_ = 64 μg/mL). The result of MIC_50_ was reasonably consistent with those of previous studies [13,24]. As a novel pleuromutilin derivative, GML-12 showed more effective antibacterial activity against *S. suis* strains (MIC_50_ = 0.125 μg/mL). In the checkerboard assay, the TET–GML-12 interactions against the S40, SNJ-5, S11 and ATCC 43,765 strains were mainly synergistic, while those against the other *S. suis* strains were additive. In the time-killing assay, compared with the most active drug alone, the TET/GML-12 combination had a synergetic effect and an additive effect against three strains and the S40 strain, respectively. The combined effect of the TET/GML-12 combination against four *S. suis* strains in time-killing assays was similar to that of the chessboard assays. However, when against SNJ-5 and S40, the most effective doses of the TET/GML-12 combination (TET at 1 × MIC plus GML-12 at 2 × MIC against SNJ-5; TET at 2 × MIC plus GML-12 at 2 × MIC against S40) obtained by the time-killing assay were higher than those obtained by the checkerboard test. The reason for the difference might be that the time-killing assay records the reduction in viable bacteria, while the checkerboard test measures turbidity that can be due to viable (dividing or not) and dead bacteria. [29,30]. Interestingly, the growth curve of the control group showed a downward trend between 9 and 24 h, which may be caused by the nutritional deficiency and toxicity of metabolites [31].

A study showed that the combination of doxycycline (a TET derivative) and pleuromutilins displayed a synergetic effect against *Acinetobacter baumannii* in vitro and in a mouse infection model [20]. In our study, the synergetic effects between TET and GML-12 in both the checkerboard and time-killing assays motived us to evaluate the effect of the TET/GML-12 combination against *S. suis* in vivo. The in vivo antibacterial activity of the combined therapy was further confirmed by the mouse thigh infection model. The thigh infection level of these four strains in the untreated group suggested that the *S. suis* strains successfully colonized in the mouse thigh, and their bacterial density ranged from 8.78 to 10.18 log_10_ CFU/g. After 24 h of treatment, differences of 2.0–4.6 log_10_ CFU/g (*p* ≤ 0.0016) were achieved between the untreated groups and the combined treatment groups. The TET/GML-12 combination exerted a synergistic effect against ATCC 43765, S40 and SNJ-5.

Unexpectedly, the TET/GML-12 combination against S11 showed an indifferent effect in the mouse thigh infection model but indicated a synergistic effect in the time-killing assay and checkerboard test. Compared with in vivo experiments, in vitro experiments lack complex metabolic processes, so the results of these two are not always consistent [32]. Additionally, the bacterial density in all GML-12 monotherapy groups was lower than that in untreated groups. Although the dose of GML-12 was much less than that of TET, its antibacterial activity was still better than or similar to that of TET. In the acute toxicity experiment, GML-12 at 20 mg/kg was safe for oral and intravenous administration in mice. Given its high antibacterial efficiency and low toxicity, GML-12 preliminarily showed potential use in internal medicine.

However, there were some deficiencies in this study. Only four *S. suis* strains had been tested in time-killing assays and thigh infection models, which might not fully understand the effect of the GML-12/TET combination. Although the data of the thigh infection model provided preliminary guidance for reasonable combined therapy, there were physiological differences among different species and the results of the thigh infection model may not be directly used for clinical treatment.

In this study, when TET was combined with GML-12, the therapeutic effect was achieved by a lower dose of TET. Such a strategy may also provide an option to significantly reduce the dose without affecting the effectiveness of the treatment, which in turn may reduce the side effects associated with the use of TET. It was especially important when prolonged therapy was required. A combination of antibiotics may help slow down the evolution of resistance and increase antimicrobial efficacy [33]. We believe that the GML-12/TET combination therapy might restore the TET sensitivity of other TET-resistant bacteria.

## 4. Materials and Methods

### 4.1. Chemicals

GML-12 (purity: 98%) (22-(4-(2-(4-Nitrophenyl-piperazin-1-yl)-acetyl)-piperazin-1-yl)-22-deoxypleuromutilin) was synthesized according to our previous work [22]. Tetracycline (TET) was purchased from Guangzhou Xiang Bo Biological Technology Co., Ltd. (Guangzhou, China). All other reagents and solvents were obtained from Becton, Dickinson and Company (Franklin Lake, New Jersey, America) and were used as supplied.

### 4.2. Microorganisms

*S. suis* ATCC 43,765 strains were obtained from China General Microbiological Culture Collection Center (Beijing, China). The remaining eleven clinical strains (SNJ-5, S40, S11, S1-2, S94, S110, S75, S12, S114, S2013-1025 and S41) were originally isolated from pigs and saved in our laboratory at present.

### 4.3. Animal

Six-week-old female specific pathogen-free (SPF) ICR mice were purchased from the Hunan SJA Laboratory Animal Co., Ltd (Hu Nan, China) under license number SCXK2011-0003. All mice were raised and maintained at the Laboratory Animal Center of South China Agricultural University. All animal experiments were in line with the Ethical principles of Animal Research and have been approved by the Ethics Committee of Guangdong Medical Experimental Animal Center.

### 4.4. In Vitro Antibacterial Activity

#### 4.4.1. Determination of Minimum Inhibitory Concentration

The MIC values of GML-12 and TET against 12 isolates were determined by a broth microdilution method which was according to the Clinical and Laboratory Standards Institute (CLSI) guidelines [34]. The MIC results of tetracycline against *S. suis* strains were classified as sensitive, intermediate and resistant using CLSI available breakpoint.

#### 4.4.2. Checkerboard Microdilution Assay

To analyze the possible interaction between GML-12 and TET, the checkerboard method was performed [35]. In brief, 96-well plates containing eight serial 2-fold dilutions of TET + GML-12 (range, 0.25 × MIC to 16 × MIC) were added to the bacterial suspension (10^5^ CFU/mL) and incubated for 16 h at 37 °C. Control wells were free of antimicrobial substances. All experiments were conducted at least three times. The fractional inhibitory concentration index (FICI) was calculated using the following equation: FICI = [(MIC of GML-12 in combination)/(MIC of GML-12 alone)] + [(MIC of tetracycline in combination)/(MIC of tetracycline alone)]. The sum of each FICI was interpreted as follows: FICI ≤ 0.5 suggests synergy; 0.5 < FICI ≤ 1 suggests additivity; 1< FICI ≤4 suggests indifference; 4 < FICI indicates an antagonistic effect [36].

#### 4.4.3. Time-Killing Assay

To further study the antibacterial effect of TET combined with GML-12, we evaluated the in vitro killing activities of two drugs against *S. suis*. For bacterial inoculation, four strains were selected according to the most dominant FICI. The strains were grown in Trypticase soy broth (TSB) containing 5% FBS at the starting inoculum of 10^6^ CFU/mL. The inoculum in TSB was used to determine the time-kill curves of GML-12 (1×, 2× MIC) and TET (1×, 2× MIC) alone, and in combination in the tubes. TSB without antibiotics was used as the control. All tubes were cultured in an incubator-shaker at 37 °C (150 rpm). At 0, 3, 6, 9 and 24 h of incubation, 100-μL aliquots were removed from each tube. The aliquots were serially diluted and inoculated onto Trypticase Soy Agar (TSA) plates. All plates were incubated at 37 °C (22–24 h) for counting colonies.

The results of all experiments were performed in triplicate and indicated as logarithms with corresponding standard errors. If the bacterial count (CFU/mL) reduced by ≥ 2 log_10_ for the combination therapy in comparison to the most active monotherapy, the interaction was identified as synergistic [37]. A bactericidal effect was defined as a ≥ 3 log_10_ increase in killing within 24 h [38]. Additivity was defined as a 1 log_10_ to 2 log_10_ increase in killing with the combination therapy in comparison with the most active monotherapy. Indifference was interpreted as a < 1 log_10_ CFU/mL killing or growth [39].

### 4.5. In Vivo Assessment

#### 4.5.1. In Vivo Toxicity

The acute toxicity of GML-12 was carried out in mice according previous reports [40]. First of all, the GML-12 powder was fully dissolved in the solvent, which was composed of 5% dimethyl sulfoxide (DMSO), 5% Tween 80 and 90% normal saline. A total of 100 female ICR mice were randomly divided into five groups (each with twenty mice) and injected with GML-12 solution and the solvent, respectively, via a single oral or intravenous administration route. The GML-12 was administered at a high dose of 20 mg/kg. Untreated mice served as a control group. The change in body weight and clinical signs were monitored for two weeks following treatment.

#### 4.5.2. In Vivo Anti-*S. suis* Activity

The mice thigh infection model reported in previous work [41] was used to evaluate the synergistic effect of GML-12 and TET in combination. The mice were rendered neutropenia (neutrophil count < 100 mm^3^) with 150 and 100 mg/kg intraperitoneal injections of cyclophosphamide (CPM), administered 4 and 1 days before bacterial inoculation, respectively. Each thigh of mice was inoculated intramuscularly (i.m.) with 100μL of inoculum including 10^6^ log phase of test strains. Mice were treated with TET and GML-12 alone or in combination at 2 h after inoculation. TET using subcutaneous (SC) injection and GML-12 using intraperitoneal injection. Mice in the untreated control group were injected with normal saline only. After 22 h of antibiotic treatment, animals were humanely sacrificed by cervical dislocation. Both posterior thigh muscles of mice were immediately collected in normal saline and diluted properly after being homogenized to determine CFU counts by the flat colony counting method.

The thigh tissue CFU titer was indicated as log_10_ CFU/g. Data were analyzed by an unpaired, two-tailed Student’s t-test with the SPSS software (19.0.1). Only when the results showed a statistically significant decrease in CFU counts (*p* < 0.05) did the antibiotic regimen (monotherapy or combination therapy) suggest efficacy compared to other regimens or untreated groups. The combination was considered to have a synergistic effect when the combined treatment induced a higher reduction in colony counts than the monotherapy. A combination was considered to be antagonistic when the combination caused a lower reduction in colony counts yielded than with monotherapy [42]. Moreover, CFU reductions of more than 3 log_10_ indicated a bactericidal effect [43].

## 5. Conclusions

In this study, high-level resistance to TET and favorable sensitivity to GML-12 were found in 12 *S. suis* strains. In the chessboard assay, the TET/GML-12 combination against 12 *S. suis* isolates showed synergistic or additive effects. These synergistic interactions were further confirmed by time-killing assays and mouse thigh infection models. The time-killing assay verified the synergistic effect of the TET/GML-12 combination against ATCC 43,765 and S11 and the additive effect against S40. In mouse thigh infection models, the TET/GML-12 combination exerted a synergistic effect against ATCC 43765, S40 and SNJ-5. There was no antagonism in all experiments. These results also suggest that the GML-12/TET combination is effective in reducing the antibiotic dose for *S. suis* treatment. Collectively, the GML-12/TET combination therapy offered the potential to revive the potency of TET as a conventional antibiotic against *S. suis* strains.

## Figures and Tables

**Figure 1 molecules-25-03522-f001:**
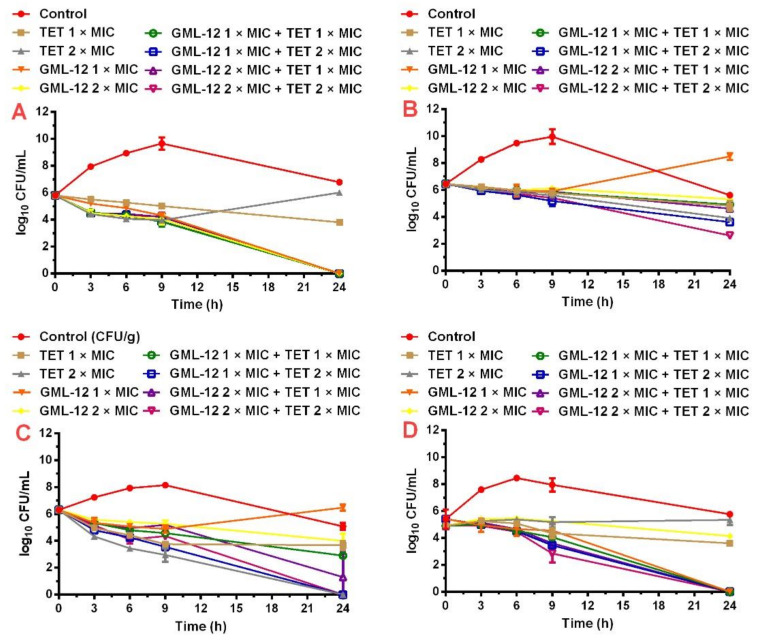
Time-kill curves of TET and GML-12 alone and in combination against ATCC 43,765 (**A**), S40 (**B**), SNJ-5 (**C**) and S11 (**D**). The mean of three biological replicates is shown and error bars represent the s.d.

**Figure 2 molecules-25-03522-f002:**
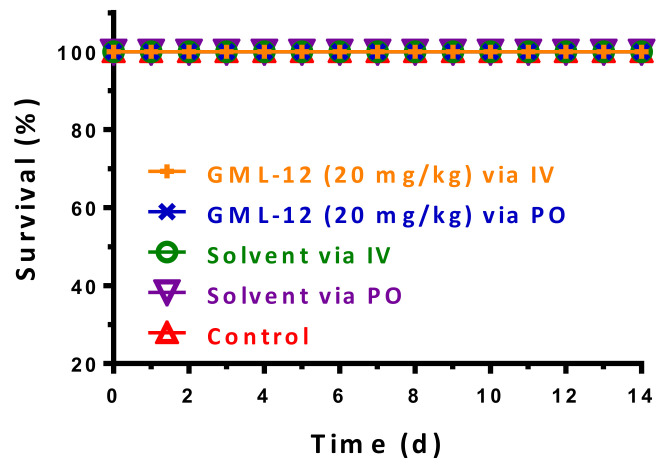
Acute toxicity of GML-12 to mice by intravenous (IV) and oral (PO) routes of administration.

**Figure 3 molecules-25-03522-f003:**
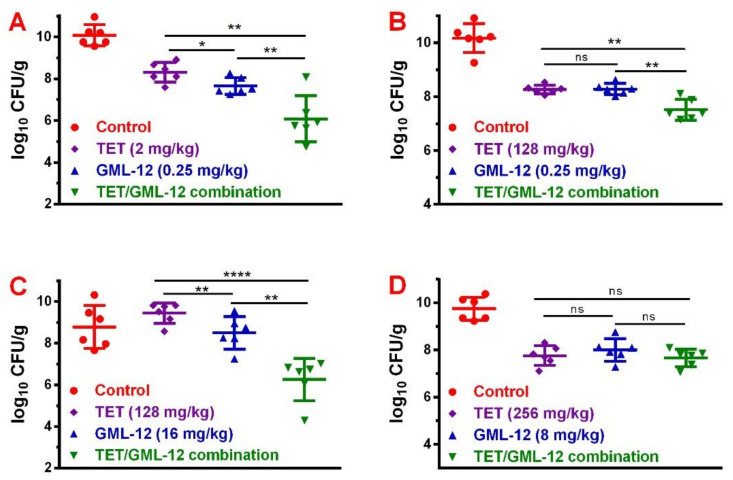
Therapeutic effect of TET and GML-12 alone and in combination against ATCC 43,765 (**A**), S40 (**B**), SNJ-5 (**C**) and S11 (**D**) in neutropenic mice. The bacterial load of infected thigh muscle in neutropenic mice (*n*= 6 per group) was determined by colony counting. All data are mean ± s.d. *p* values were determined using an unpaired, two-tailed Student’s t-test. ns, *p* > 0.05; *, *p* < 0.05; **, *p* < 0.01; ***, *p* < 0.001; ****, *p* < 0.0001.

**Table 1 molecules-25-03522-t001:** Summary of MIC values of GML-12 and TET, alone and in combination, against *S. suis* strains.

Strains	MIC (μg/mL) Along		MIC (μg/mL) Combined			
	TET (Clinical Breakpoints ^a^)	GML-12	TET	GML-12	FICI ^b^	Interpretation
**ATCC 43765**	1 (I)	0.125	0.25	0.03125	0.5	synergistic
**S2013-1025**	64 (R)	8	32	4	1	additivity
**S94**	64 (R)	0.125	32	0.0625	1	additivity
**S41**	128 (R)	4	32	2	0.75	additivity
**S110**	32 (R)	0.25	4	0.125	0.625	additivity
**S114**	64 (R)	0.125	8	0.0625	0.625	additivity
**S40**	64 (R)	0.25	16	0.0625	0.5	synergistic
**S1-2**	0.25 (S)	0.0625	0.0625	0.03125	0.75	additivity
**S11**	128 (R)	4	32	1	0.5	synergistic
**S12**	64 (R)	0.25	16	0.125	0.75	additivity
**S75**	64 (R)	0.25	32	0.125	1	additivity
**SNJ-5**	64 (R)	8	8	2	0.375	synergistic

^a^ Clinical breakpoints were obtained from the Clinical and Laboratory Standards Institute standards. S: susceptible, I: intermediate, R: resistant (R ≥ 2μg/mL; 0.5 μg/mL < I < 2 μg/mL; S ≤ 0.5 μg/mL) [24,25]. ^b^ FICI ≤ 0.5 synergy; 0.5 < FICI ≤ 1, additivity; 1 < FICI ≤ 4, indifference; 4.0 < FICI, antagonism.

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
