# Peer review of "Synergistic Effect of a Pleuromutilin Derivative with Tetracycline against Streptococcus suis In Vitro and in the Neutropenic Thigh Infection Model"

_molecules, 2020, doi:10.3390/molecules25153522_

Round 1

Reviewer 1 Report

General comments

The quality of the language should be carefully reviewed.

As describe in the instructions for the authors, “Abbreviations should be defined in parentheses the first time they appear in the abstract, main text, and in figure or table captions and used consistently thereafter”. So please do so.

Please review carefully the text to properly write S. suis. There is a space between the period and the s of suis…

Bacterial name should always be italicized, please carefully review the manuscript.

Sometimes in vivo is italicized sometimes not... Please review the manuscript.

Please provide more information in the introduction about pleuromutilin

Statistical analysis should also be carefully reviewed (please see comments below).

Specific comments :

Abstract

Please present the context of this study at the beginning of the abstract

Introduction

L41: Please review this sentence for the quality of the language.

L44-45 : Please review this sentence

L49-50: Please review this sentence for the quality of the language

Materials and Methods

Were the in vivo studies performed according to animal care guidelines ? If yes, please precise.

Please clarify how the statistical analysis was performed. Why paired sample T test was selected although samples seem independent (not paired). How comparison between the monotherapy and combination are performed ?

Discussion

L142: Please remove the parentheses

L147-148: To be more precise: time killing assay measure viable bacteria while checkerboard measure turbidity that can be due to viable (dividing or not) and dead bacteria.

L151-154: Please split this sentence in 2. The first part is about A. baumanii while the second par is about S. suis. Dividing the sentence would make it clearer and, I would suggest to add S. suis in the 2 sentences to increase clarity.

L159: suggestion; remove “in all thigh infection models”

L161-166: This should be in the introduction section

L169-170: Please review this sentence.

L172-173: “Its antibacterial (GML-12) activity was still better or similar to that of TET.” Have you done any statistical analysis to state that it is better or  are you observing that with the graph? Please be more precise.

Figure 1: please use the same scale for all the graphs for their y axis (0-12) to help visual comparison.

Figure 3:  Figures should be self-explanatory, please provide more information in the legend (What are the vertical and horizontal bars standing for ? What statistical analysis was performed ? What does the **** or ** mean ?

Author Response

Dear Editor,

Thank you very much for managing and coordinating the review process for our manuscript entitled “Synergistic effect of a pleuromutilin derivative with tetracycline against Streptococcus suis in vitro and in the neutropenic thigh infection model” (moleculars-865038). We appreciate the valuable comments from the reviewers. We have completed the corrections and revised the manuscript as requested by the reviewers, accordingly. We hope that these revisions and replies to the reviewers’ comments will strengthen the manuscript to meet the high quality of publication standards of the molecules.

All changes are highlighted in red in the “Revised Manuscript”, and our responses to the comments are itemized under “Responses to Referees” for your consideration of publication in molecules. If you have any questions regarding to our manuscript, please contact us at any time.

We are looking forward to your further kind considerations.

Thank you again for your help and suggestions.

Best wishes,

Yours sincerely,

You-Zhi Tang

Guangdong Provincial Key Laboratory of Veterinary Pharmaceutics Development and Safety Evaluation,

College of Veterinary Medicine, South China Agricultural University,

Add: No. 483 Wushan Road, Tianhe District, Guangzhou, Guangdong, P.R. China

Zip: 510642

Tel: +86-020-85280665

Mobile: +86-015899976723

E-mail: youzhitang@scau.edu.cn

According with your advice, we amended the relevant part in manuscript. Some of your questions were answered below.

  • The quality of the language should be carefully reviewed.

Reply: Thank you for your advice. The language and grammar were checked in detail and corrections had been made.

  • As describe in the instructions for the authors, “Abbreviations should be defined in parentheses the first time they appear in the abstract, main text, and in figure or table captions and used consistently thereafter”. So please do so.

Reply: Many thanks the reviewer’s comments. All the abbreviations were defined in parentheses the first time they appear in the abstract, main text, and in figure or table captions and used consistently thereafter.

  • Please review carefully the text to properly write S. suis. There is a space between the period and the s of suis…

Reply: Many thanks the reviewer’s comments. All the “S.suis” in this manuscript were changed to “S. suis”.

  • Bacterial name should always be italicized, please carefully review the manuscript.

Reply: Many thanks the reviewer’s comments. All the bacterial name in this manuscript were changed to italics.

  • Sometimes in vivo is italicized sometimes not... Please review the manuscript.

Reply: Many thanks the reviewer’s comments. All the “in vivo” and “in vitro” in this manuscript were modified to be italicized.

  • Please provide more information in the introduction about pleuromutilin

Reply: Thank you for your advice. More introduction about pleuromutilin were added to Lines 54-56: It is well established that the antibacterial mechanisms of pleuromutilin derivatives are to inhibit protein biosynthesis by blocking the peptidyl transferase center (PTC) of the 50S ribosomal protein L3 [21]. GML-12 was also found to form a hydrogen bond with 50s ribosome in the previous docking study.

Statistical analysis should also be carefully reviewed (please see comments below).

Specific comments :

  • Abstract
  1. Please present the context of this study at the beginning of the abstract

Reply: Many thanks the reviewer’s comments. The context of this study was added to L15 of the abstract. L15: Tetracycline (TET) has been widely used in the treatment of Streptococcus suis (S. suis) infection. However, it was found that the sensitivity of many antibiotics in S. suis decreased significantly, especially tetracycline.

  • Introduction
  1. L41: Please review this sentence for the quality of the language.

Reply: Many thanks the reviewer’s comments. L41 was changed to L43-44: Tetracycline (TET), as a broad-spectrum antibiotic, is widely used all over the world due to its efficacy, easily absorbed, cheap and minimum side effects [12].

  1. L44-45 : Please review this sentence

Reply: Many thanks the reviewer’s comments. L44-45 were changed to L46-48 in the revised manuscript:Although the development of antibiotics can meet the challenge of antibiotic resistance, there are some problems still existing such as it takes a long time and a high cost to bring new drugs to the clinical use.

  1. L49-50: Please review this sentence for the quality of the language

Reply: Many thanks the reviewer’s comments. L49-50 were changed to L51-52: Restoring the sensitivity of previously resistant bacteria to old antibiotics may be the most powerful effect of a synergistic combination [17, 19].

  • Materials and Methods
  1. Were the in vivo studies performed according to animal care guidelines? If yes, please precise.

Reply: Thank you for your advice. We have added it in more detail.

L212-213: All animal experiments were in line with the Ethical principles of Animal Research and have been approved by the Ethics Committee of Guangdong Medical Experimental Animal Center.

  1. Please clarify how the statistical analysis was performed. Why paired sample T test was selected although samples seem independent (not paired). How comparison between the monotherapy and combination are performed ?

Reply: Many thanks the reviewer’s comments. Data were analyzed by unpaired two-tailed Student’s t-test with SPSS software in L267-268 in the revised manuscript.

  • Discussion
  1. L142: Please remove the parentheses

Reply: Thank you for your advice. The parentheses was removed in the revised manuscript.

L155: TET/GML-12 combination had a synergetic effect and an additive effect against three strains and S40 strain, respectively.

  1. L147-148: To be more precise: time killing assay measure viable bacteria while checkerboard measure turbidity that can be due to viable (dividing or not) and dead bacteria.

Reply: Thank you very much for your advice.

L159-161: The reason for the difference might be that the time-killing assay records the reduction of viable bacteria, while the checkerboard test measures turbidity that can be due to viable (dividing or not) and dead bacteria.

  1. L151-154: Please split this sentence in 2. The first part is about A. baumanii while the second part is about S. suis. Dividing the sentence would make it clearer and, I would suggest to add S. suis in the 2 sentences to increase clarity.

Reply: Thank you for your advice. The sentence was divided into two parts.

L164-167: A study showed that the combination of doxycycline (a TET derivative) and pleuromutilins displayed a synergetic effect against Acinetobacter baumannii in vitro and in a mouse infection model of infection [20]. In our study, the synergetic effects between TET and GML-12 in both checkerboard and time-killing assays motived us to evaluate the effect of TET/GML-12 combination against S. suis in vivo.

  1. L159: suggestion; remove “in all thigh infection models”

Reply: Thank you for your advice. The “in all thigh infection models” was removed.

  1. L161-166: This should be in the introduction section

Reply: Thank you for your advice. We removed it to the introduction section. L55-61.

  1. L169-170: Please review this sentence.

Reply: Thank you for your advice. L169-170 were changed to L176-177: The difference may have resulted from the in-vitro study did not simulate the actual internal environment of organisms.

  1. L172-173: “Its antibacterial (GML-12) activity was still better or similar to that of TET.” Have you done any statistical analysis to state that it is better or are you observing that with the graph? Please be more precise.

Reply: Thank you for your advice. We added the significance analysis between the two groups in fig. 3 in the revised manuscript.

Figure 1: please use the same scale for all the graphs for their y axis (0-12) to help visual comparison.

Reply: Thank you for your advice. The image format has been changed (fig.1). And more specific illustrations were added: The mean of three biological replicates was shown and error bars represent the s.d.

Figure 3:  Figures should be self-explanatory, please provide more information in the legend (What are the vertical and horizontal bars standing for ? What statistical analysis was performed ? What does the **** or ** mean ?

Reply: Thank you for your advice. The statistical analysis and the P vaules were added in Fig.3, L122-125: The bacterial load of infected thigh muscle in neutropenic mice (n= 6 per group) was determined by colony counting. All data are mean ± s.d. P values were determined using an unpaired, two-tailed Student’s t-test. ns, P > 0.05; *, P < 0.05; **, P < 0.01; ***, P <0.001; ****, P < 0.0001.

Reviewer 2 Report

Manuscript by Fang Chen et al. “Synergistic effect of a pleuromutilin derivative with 3 tetracycline against Streptococcus Suis in vitro and in 4 the neutropenic thigh infection model” focused on the effect of pleuromutilin derivative (GML-12) in combination with tetracycline (TEM) to against 12 Streptococcus suis isolates. The authors analyzed the possibilities of introducing a new treatment against potentially pathogenic bacteria showing increasingly frequent resistance to previously used drugs. The results are interesting and deserve attention. Although the TET/GML-12 combination exhibited synergistic and additive effects against S. suis, the number of strains does not guarantee the correctness and repeatability of results. As the authors themselves have stated, only four S. suis strains had been tested in time-killing assays and thigh infection models, which might not fully understand the effect of the GML-12/TET combination. This, in my opinion, does not allow to unequivocally draw conclusions from research carried out by Fang Chen et al. In order to allow the publication of the manuscript for publication, I recommend expanding the research with more strains.

Minor Comments:

  • In the title of the publication please change "Suis" to "suis".
  • Please add spaces in the "S. suis" in the entire manuscript.
  • Line 50: After the word "against" please add a comma.
  • Line 101: In the title of subsection 2.2.1, please correct "CML-12" to "GML-12".

Author Response

Dear Editor,

Thank you very much for managing and coordinating the review process for our manuscript entitled “Synergistic effect of a pleuromutilin derivative with tetracycline against Streptococcus suis in vitro and in the neutropenic thigh infection model” (EJMECH-D-20-00749). We appreciate the valuable comments from the reviewers. We have completed the corrections and revised the manuscript as requested by the reviewers, accordingly. We hope that these revisions and replies to the reviewers’ comments will strengthen the manuscript to meet the high quality of publication standards of the molecules.

All changes are highlighted in red in the “Revised Manuscript”, and our responses to the comments are itemized under “Responses to Referees” for your consideration of publication in molecules. If you have any questions regarding to our manuscript, please contact us at any time.

We are looking forward to your further kind considerations.

Thank you again for your help and suggestions.

Best wishes,

Yours sincerely,

You-Zhi Tang

Guangdong Provincial Key Laboratory of Veterinary Pharmaceutics Development and Safety Evaluation,

College of Veterinary Medicine, South China Agricultural University,

Add: No. 483 Wushan Road, Tianhe District, Guangzhou, Guangdong, P.R. China

Zip: 510642

Tel: +86-020-85280665

Mobile: +86-015899976723

E-mail: youzhitang@scau.edu.cn

Comments and Suggestions for Authors

Manuscript by Fang Chen et al. “Synergistic effect of a pleuromutilin derivative with 3 tetracycline against Streptococcus Suis in vitro and in 4 the neutropenic thigh infection model” focused on the effect of pleuromutilin derivative (GML-12) in combination with tetracycline (TEM) to against 12 Streptococcus suis isolates. The authors analyzed the possibilities of introducing a new treatment against potentially pathogenic bacteria showing increasingly frequent resistance to previously used drugs. The results are interesting and deserve attention. Although the TET/GML-12 combination exhibited synergistic and additive effects against S. suis, the number of strains does not guarantee the correctness and repeatability of results. As the authors themselves have stated, only four S. suis strains had been tested in time-killing assays and thigh infection models, which might not fully understand the effect of the GML-12/TET combination. This, in my opinion, does not allow to unequivocally draw conclusions from research carried out by Fang Chen et al. In order to allow the publication of the manuscript for publication, I recommend expanding the research with more strains.

Reply:

Many thanks the reviewer’s comments. It is necessary to test multiple strains to obtain conclusions. In this manusript, although time-killing assays and thigh infection models tested only 4 strains of S. suis, this also has a specific indicative significance. In many studies about the combination effect, 1 ~ 6 strains of bacteria were often used for experiments in vivo and in vitro. The references are as follows:

  1. The article “A New Combination of a Pleuromutilin Derivative and Doxycycline for Treatment of Multidrug-Resistant Acinetobacter baumannii” (DOI 10.1021/acs.jmedchem.6b01805): Only the baumannii (ATCC19606) was used for in vivo experiments.
  2. The paper “In vitro synergistic activities of cefazolin and nisin A against mastitis pathogens” (DOI 10.1292/jvms.17-0180): Only six strains of mastitis pathogens were used for time-killing assays to demonstrate the synergistic effect.
  3. The paper “Synergistic efficacy of meropenem and rifampicin in a murine model of sepsis caused by multidrug-resistant Acinetobacter baumannii” (DOI 10.1016/j.ejphar.2014.02.015): Six Acinetobacter baumannii isolates were used for time-killing assays, and only one of these strains was used for the murine model assay.
  4. The article “In vitro and in vivo activity of single and dual antimicrobial agents against KPC-producing Klebsiella pneumoniae” (DOI 10.1093/jac/dkx419): Four strains were performed in the time-kill assay and G. mellonella killing studies.

There are many similar examples.

However, there is no doubt that the persuasive conclusion is drawn from a large number of experiments. We will pay more attention to this problem in future experiments. Thank you for your valuable advice.

Minor Comments:

  • In the title of the publication please change "Suis" to "suis".
  • Reply: Thank you for your comments. "Suis" was changed to "suis" in the title.
  • Please add spaces in the "S. suis" in the entire manuscript.

Reply: Thank you for your comments. All the “S.suis” were changed to “S. suis

  • Line 50: After the word "against" please add a comma.

Reply: Thank you for your comments. We have modified this sentence.

L51-L52: Restoring the sensitivity of previously resistant bacteria to old antibiotics may be the most powerful effect of a synergistic combination.

  • Line 101: In the title of subsection 2.2.1, please correct "CML-12" to "GML-12".

Reply: Thank you for your comments. We have corrected it in L110 in the revised manuscript.

Round 2

Reviewer 1 Report

General comments

Thanks you for the revision of the manuscript. Most of my questions/ comments were addressed. However there are still few points that could be improved.

Specific comments :

Abstract

L16 : Suggestion: the efficacy of many antibiotics against S. suis decreased

L21 Please define FICI

Introduction

L31 : leave the beginning of the sentence as it was : Streptococcus suis (S. suis)

L31: that causes

L43: I would suggest: due to its efficacy, its easy absorption, its low price and its minimum side effects

L59 and 60: 50S

L61: 30S

Discussion

L188-192: This is still not clear. Please review it again.

Author Response

Dear Editor,

Thank you very much for managing and coordinating the review process for our manuscript entitled “Synergistic effect of a pleuromutilin derivative with tetracycline against Streptococcus suis in vitro and in the neutropenic thigh infection model” (moleculars-865038). We appreciate the valuable comments from the reviewers. We have completed the corrections and revised the manuscript as requested by the reviewers, accordingly. We hope that these revisions and replies to the reviewers’ comments will strengthen the manuscript to meet the high quality of publication standards of the molecules.

All changes are highlighted in red in the “Revised Manuscript”, and our responses to the comments are itemized under “Responses to Referees” for your consideration of publication in molecules. If you have any questions regarding to our manuscript, please contact us at any time.

We are looking forward to your further kind considerations.

Thank you again for your help and suggestions.

Best wishes,

Yours sincerely,

You-Zhi Tang

Guangdong Provincial Key Laboratory of Veterinary Pharmaceutics Development and Safety Evaluation,

College of Veterinary Medicine, South China Agricultural University,

Add: No. 483 Wushan Road, Tianhe District, Guangzhou, Guangdong, P.R. China

Zip: 510642

Tel: +86-020-85280665

Mobile: +86-015899976723

E-mail: youzhitang@scau.edu.cn

According with your advice, we amended the relevant part in manuscript. Some of your questions were answered below.

In the manuscript we used annotations to highlight your comments and used a different color font than last time.

Specific comments :

Abstract

  • L16 : Suggestion: the efficacy of many antibiotics against S. suis decreased

Reply:Thank you for your advice. L16: “the sensitivity of many antibiotics” was changed to “the efficacy of many antibiotics”

  • L21 Please define FICI

Reply:Many thanks the reviewer’s comments. We've added the exact value of FICI: In-vitro time-killing assays and in-vivo therapeutic experiments were used to confirm the synergistic effect of TET/GML-12 combination against S. suis strains screened based on the FICI ≤ 0.5.

Introduction

  • L31 : leave the beginning of the sentence as it was : Streptococcus suis (S. suis)

Reply: Many thanks the reviewer’s comments. L31: Streptococcus suis (S. suis)

  • L31: that causes

Reply: Many thanks the reviewer’s comments. “that cause” was changed to “that causes”.

  • L43: I would suggest: due to its efficacy, its easy absorption, its low price and its minimum side effects

Reply: Thank you for your advice. L43: Tetracycline (TET), as a broad-spectrum antibiotic, is widely used all over the world due to its easy absorption, its low price and its minimum side effects.

  • L59 and 60: 50S

Reply: Many thanks the reviewer’s comments. L60 and 61: “50S” was changed to “50S”

  • L61: 30S

Reply: Many thanks the reviewer’s comments. L62: “30S” was changed to “30S”

Discussion

  • L188-192: This is still not clear. Please review it again.

Reply: Many thanks the reviewer’s comments. L188-192 was changed to L189-190: Compared with in vivo experiments, in vitro experiments lack complex metabolic processes, so the results of these two are not always consistent.

Reviewer 2 Report

The authors have addressed the reviewer comments. The paper should be accepted.

Author Response

Dear Editor,

Thank you for your approval.

If you have any questions regarding to our manuscript, please contact us at any time.

Best wishes,

Yours sincerely,

You-Zhi Tang

Guangdong Provincial Key Laboratory of Veterinary Pharmaceutics Development and Safety Evaluation,

College of Veterinary Medicine, South China Agricultural University,

Add: No. 483 Wushan Road, Tianhe District, Guangzhou, Guangdong, P.R. China

Zip: 510642

Tel: +86-020-85280665

Mobile: +86-015899976723

E-mail: youzhitang@scau.edu.cn